# Hopes, joys and fears: Meaning and perceptions of viral load testing and low-level viraemia among people on antiretroviral therapy in Uganda: A qualitative study

Nicholus Nanyeenya[1,2]*, Godfrey Siu[3], Noah Kiwanuka[1], Fredrick Makumbi[1], Esther Nasuuna[4], Damalie Nakanjako[5], Gertrude Nakigozi[6], Susan Nabadda[2], Charles Kiyaga[2], Simon P. S. Kibira[7]

1 Department of Epidemiology and Biostatistics, School of Public Health, Makerere University College of Health Sciences, Kampala, Uganda, 2 Ministry of Health Central Public Health Laboratories, Kampala Uganda, 3 Child Health and Development Centre, School of Medicine Makerere University College of Health Sciences, Kampala, Uganda, 4 Infectious Diseases Institute, Makerere University College of Health Sciences, Kampala, Uganda, 5 Department of Medicine, School of Medicine Makerere University College of Health Sciences, Kampala, Uganda, 6 Rakai Health Sciences Project, Rakai, Uganda, 7 Department of Community Health and Behavioral Sciences, School of Public Health, Makerere University College of Health Sciences, Kampala, Uganda

* nanyeenya@gmail.com

## Abstract

Uganda applies the World Health Organization threshold of 1,000 copies/ml to determine HIV viral non-suppression. While there is an emerging concern of low-level viraemia ($\geq$50 to <1,000 copies/ml), there is limited understanding of how people on antiretroviral therapy perceive viral load testing and low-level viremia in resource-limited settings. This qualitative study used the health belief model to explore the meaning that people living with HIV attach to viral load testing and low-level viraemia in Uganda. We used stratified purposive sampling to select people on antiretroviral therapy from eight high volume health facilities from the Central, Eastern, Northern and Western regions of Uganda. We used an interview guide, based on the health belief model, to conduct 32 in-depth interviews, which were audio-recorded and transcribed verbatim. Thematic analysis technique was used to analyze the data with the help of ATLAS.ti 6. The descriptions of viral load testing used by the participants nearly matched the medical meaning, and many people living with HIV understood what viral load testing was. Perceived benefits for viral load testing were the ability to show; the amount of HIV in the body, how the people living with HIV take their drugs, whether the drugs are working, and also guide the next treatments steps for the patients. Participants reported HIV stigma, lack of transport, lack of awareness for viral load testing, delayed and missing viral load results and few health workers as the main barriers to viral load testing. On the contrary, most participants did not know what low-level viraemia meant, while several perceived it as having a reduced viral load that is suppressed. Many people living with HIV are unaware about low-level viraemia, and hence do not understand its associated risks. Likewise, some people living with HIV are still not aware about viral load testing. Lack of transport, HIV stigma and delayed viral load results are major barriers to viral load testing.

**Data Availability Statement:** All relevant data are within the paper and its Supporting Information files.

**Funding:** Research reported in this publication was supported by the Fogarty International Center, National Institute of Alcohol Abuse and Alcoholism, National Institute of Mental Health, of the National Institutes of Health under Award Number D43 TW011304, (awarded to NN, through The Makerere University Behavioral and Social Science Research program). The content is solely the responsibility of the authors and does not necessarily represent the official views of the National Institutes of Health. The funders had no role in study design, data collection and analysis, decision to publish, or preparation of the manuscript.

**Competing interests:** The authors have declared that no competing interests exist.

Hence, there is an imminent need to institute more strategies to create awareness about both low-level viraemia and viral load testing, manage HIV related stigma, and improve turn-around time for viral load results.

## Introduction

Uganda currently applies the World Health Organization (WHO) threshold of 1,000 copies/ml to determine HIV viral non-suppression [1], and this indicates either poor drug adherence or virological failure. People living with HIV (PWH) having a viral load (VL) of at least 50 copies/ml but less than 1,000 copies/ml have low-level viraemia [2]. Previous studies have indicated that low-level viraemia (LLV) is associated with HIV virologic failure, transmission and drug resistance [3–9]. LLV is an emerging concern in Uganda and other Sub-Saharan African (SSA) countries [10] as it threatens the efforts to achieve the global targets of ending HIV as an epidemic, as stipulated by the third Sustainable Development Goal, target 3.3 [11, 12].

Following the 2013 WHO recommendation [13], Uganda initiated her efforts to scale up HIV VL testing, to monitor the efficacy of antiretroviral therapy (ART) for all eligible PWH on ART in 2014 [14]. Since then, there has been a steady increase in the number of VL tests done annually from 16,411 tests in 2014 to 1,332,335 tests in 2020, and VL testing is now accessible in all health facilities offering ART services in Uganda [15]. The national VL testing algorithm stipulates that all PWH who have been on ART for 6 months should be offered the first VL test. If the VL results are suppressed (below 1,000 copies/ml), the VL test is repeated at 12 months, and then once every year. For non-suppressed PWH (having a VL of 1,000 copies/ml or more), they are offered monthly intensive adherence counselling (IAC) sessions for three months before a repeat VL test is done in the fourth month to determine if viral suppression has been achieved. Failure to achieve viral suppression for such a person will result into convening a switch committee which will decide on the next management steps [1].

Despite using a threshold of 1,000 copies/ml to determine VL non-suppression in many low-and-middle income countries due to its cost-effectiveness and feasibility [16], there are growing concerns that the high threshold may increase the number of PWH with LLV [10]. HIV drug resistance has been shown in PWH on ART with LLV [3], and LLV has been associated with increased risks of virologic failure [5, 6, 17]. Unlike the WHO recommendation of using 1,000 copies/ml, the International Association of Providers of AIDS Care (IAPAC) and CDC recommend the threshold of 200 copies/ml for determining HIV VL non-suppression. [18, 19].

Knowledge and positive perceptions of PWH about HIV care are very critical in achieving improved treatment outcomes, including VL suppression [20]. Although numerous studies have been conducted about the perceptions and experiences of living with HIV in Uganda [21–26], there is currently limited understanding of the meaning and perceptions that PWH attach to HIV VL testing and LLV, and the implication of this to their adherence to ART in Uganda.

With the advancing global targets to end the HIV epidemic, there is need for evidence to guide policy and service delivery decisions to improve VL testing and clinical management of low-level viraemia among PWH in Uganda. This study seeks to explore the meaning and perceptions of viral load testing and low-level viraemia among PWH on ART in Uganda.

## Methods

### Study design and setting

This was a narrative qualitative study, comprising of 32 in-depth interviews (IDIs) conducted with PWH on ART from eight high volume health facilities across all the four regions (Central,

Eastern, Northern and Western) of Uganda. From each region, one rural and one urban health facility was purposively selected for the study, having a high number of PWH with LLV, as per the national VL program data. The study findings have been reported using the consolidated criteria for reporting qualitative research (COREQ) [27].

## Participant selection

Participants were eligible for the study if they were; PWH who had been on ART for six months or more, and aged 18 years or above. They were selected by stratified purposive sampling [28], from both rural and urban settings, considering their gender, level of education, and age to obtain maximum variation. Study participants were identified at the health facility with the help of the health workers, who assessed their eligibility to participate in the study based on the set criteria, and also oriented them about the study. Only 3 participants opted not to participate in the study and these indicated that they were in a hurry, and could not take part in the study. Hence we did not include these participants in the study.

## Ethical considerations

Ethical approval was obtained from both Makerere University School of Public Health Research and Ethics Committee (Approval number is SPH-2021-144), and Uganda National Council for Science and Technology (Approval number is HS2008ES). Permission to conduct the research at the selected health facilities was obtained from the Uganda Ministry of Health. Participants were taken through an informed consent process involving understanding the study purpose, their right to voluntary participation and confidentiality. All provided written consent to participate and be audio-recorded. All methods were carried out in accordance with relevant guidelines and regulations.

## Data collection

The participants' demographic information and summarized medical history were collected using a separate form. A semi-structured interview guide, with open-ended questions developed basing on the constructs of the health belief model was used to guide the conversation exploring participant's meaning and perceptions about VL testing and low-level viremia. This interview guide was piloted with three PWH at two selected health facilities, and reviewed for appropriateness by experienced social researchers (G.S, S.P.K, and E.N) prior to its use.

The interview guide explored participants' understanding of VL testing and LLV, perceived severity of LLV and its possible clinical outcomes, how they described this condition, their adherence to treatment and VL testing to prevent LLV, and perceived susceptibility to LLV and how it related with routine VL testing and treatment adherence. Other issues were perceived benefits and barriers, and perceived competencies of PWH to undertake VL testing.

The interviews were conducted in the local languages spoken in the participants' study area (Luganda, Runyankore, Ateso or Acholi) by an experienced qualitative research team (N.N, T.J, O.S, A.J, N.S, and K.M) and each interview lasted for between 30 to 45 minutes. Daily reviews of the conducted interviews were done with the research team, and we stopped the data collection when we realized that there was no new additional data to be obtained. Audios were simultaneously transcribed and translated to English (32 interviews). Eight interviews went through the quality control process.

## Data analysis

Thematic analysis was used to analyze the data [29] with ATLAS.ti version 6.0 used to code and organise the data. Three members of the research team (N.N, E.N, and T.J) read the transcripts to familiarize themselves with the data. During this process, they marked different ideas about the data to prepare for coding. A meeting was then convened to discuss through the initial ideas generated, to guide the process of generating the initial codes. Each of the three members then started generating initial codes for the first selected set of eight transcripts. Initial codes were discussed and compared in a second meeting to obtain consensus on how to generate a codebook. All further coding was done in cyclical manner and all codes were analyzed to generate themes. We reviewed the generated themes and compared them to initial codes and the data set, to refine them.

# Results

## Characteristics of the study participants

Of the 32 PWH interviewed, 17 were female. The age range was 24 to 50 years (median age, 38.5 years), and they had been on ART for 4 to 12 years (median ART duration, 6.5 years). Slightly over half (18) of the participants were married, about one third (11) were single, while 3 were widowed. Over half of the participants were subsistence farmers. Two participants reported no formal education, while three had attained tertiary education. All the participants were on dolutegravir-based first line regimens. All were virally suppressed based on their records, with 28 having a non-detectable VL and four having low-level viraemia. The major themes identified are shown in Fig 1.

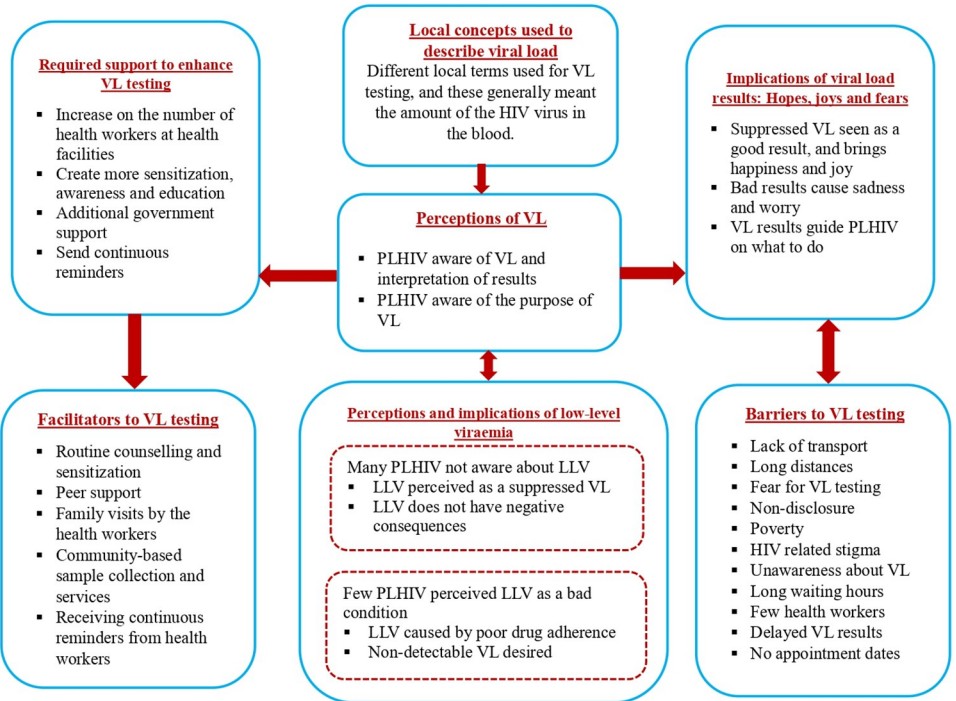

**Fig 1. Summarizing the major themes identified.**

## Viral load testing

**Local concepts used to describe viral load.**   We found that the local terms used to describe viral load in all settings matched the biomedical definition. In the central region where Luganda is the most widely spoken language, participants described viral load as '*Obungi bw'akawuka mu musaayi*' (the amount of the HIV virus in the blood). The Acholi speakers of Northern Uganda described it as '*Pimo dwong onyo nok pa kwidi twojonyo iremo*' (the number of viruses in the blood). The Eastern region Ateso speakers' phrase '*Etiai lo ekurut kotoma akuwan*', (the amount of the virus), but rather than '. . .virus in the *blood*' as in the case of the Buganda and Acholi, the Ateso phrase refers to the 'virus in the body'. Similarly for the Western region, viral load was described as '*Obwingi bw'akakooko ka sirimu omushagama*' (the amount of HIV virus in blood).

The association between VL and one's health status was very well known and discussed with relative ease. Participants' accounts provided an interesting understanding of how VL assumes relevance in the person's health and life, and highlighted both the biomedical and social importance of VL measurement. In discussing the biomedical value of VL testing, they related its function to counting the amount of disease or copies of the virus in the body.

**Perceptions of VL testing.**   Participants provided descriptions of what they perceived as viral load testing to be and its purpose. To many of the participants, viral load testing in itself did not necessarily signal that something was wrong in the body and needed to be checked. They understood that it was not always prompted by any particular symptoms, but was a necessary measure to monitor the body or blood of a person living with HIV for the level of the virus. Participants valued viral load testing as a tool for generating authentic results needed by health workers to aid their explanations of disease progress to their patient. One respondent explained.

> *Viral load is when some substances or blood samples are taken from you for testing and later, the machines will produce the test results. The health care provider is able to explain to you the amount of the virus in your blood. That is what viral load testing is.* (54 years old male from Eastern region)

Participants with secondary and tertiary education used viral load testing vocabulary, including medical terms 'suppressed' and 'non-suppressed', to interpret VL results. The participants compared the different viral load numbers to explain whether or not the virus was suppressed. Though participants with no education at all or only primary level education did not directly use the terms 'suppressed' or 'non-suppressed', they were also aware of the interpretation of VL results. All the participants knew that a VL result below 1,000 copies/ml was a good result, and that it suggested good treatment adherence and efficacy of the drugs. They were also aware that a result above 1,000 copies/ml was not desirable, and it indicated either poor drug adherence or that the drugs were not working well.

> *When VL is 1,000 or above, that means it's high and not good but when it's below 1,000, it means it is good. But when it's high, that means you are not taking your drugs well because they say that when you are not taking your drugs well, the virus will multiply in your blood daily, to even over 1,000,000.* (22 years old female from Northern region)

Participants' explanations about the purpose of VL testing could be described under four main categories; the ability of a VL test to show the amount of HIV in the body, to reveal how

the patient is taking his/her drugs, to check whether the drugs are working very well or not, and to guide the next treatments steps for the patient.

> *We were told, VL testing is how health workers are able to know the amount of the virus in your body after starting treatment and it is also how they are supposed to know if the medicine you are taking is really doing well for you or not? That is what actually they always tell us and why they remove our VL so that they see whether to continue with the same drugs, or they change to see, that the virus becomes suppressed.* (51 years old male from Central region)

The participants' description of the biomedical value of VL testing related to their understanding of adherence and the working of their ART medicines.

> *It is done to know the amount of viral copies in the blood. I don't know how to bring it, but it is to know the amount of viral copies in the blood. So, this helps us to know whether we are doing well on drugs or not since we have heard cases of drug resistance and poor adherence. So, VL process helps us to know whether we are doing well on drugs or not.* (20 years old female from Eastern region)

Routine viral load testing was almost universally accepted as a valuable component of life on ART as it acted as an important tool for decision making by health workers in the proper medical care of a patient.

> *And secondly, it can allow the doctors to know when to change the drugs for you. If the VL copies are reducing, it makes it easy for doctors to have confidence in the drug one is taking. Just a year ago, I was not doing well with my health and I asked the healthcare providers that "I take my drug very well but why is my VL high?" They recommended that they change the drug I was taking to another one and that is when my VL started coming down and I am feeling fine now.* (52 years old female from Western region)

Participants believed that VL testing was useful for health workers in diagnosing new illnesses quickly and treating them before they become complicated.

> *When the VL is done, the healthcare worker is able to tell the VL suppression status and in case there is another illness intruding, the doctor can tell that there is another illness starting in your body and it can easily be dealt with.* (42 years old female from Central region)

It was not only health workers who needed the VL test results, participants reported that they themselves used VL testing to monitor and track the progress they were making and whether their efforts to adherence to ART medication were paying off.

> *Yes, I have the capability and the right to do my VL because this is my life. There is no way I will ever say that they shouldn't test for CD4. . .no! Or that they shouldn't test for VL because I have to keep track of my VL suppression status and to determine, Am I taking my drugs well or something is wrong.* (38 years old male from Eastern region)

**Implications of viral load results: Hopes, joys and fears.** Participants attached mixed emotions to VL results. The amount of viral load found in one's body was a reminder of the gains and losses their bodies had made in the fight against the virus based on whether or not they had been adherent to the antiretroviral treatment. Thus, suppressed VL results were

described as *good results*, and these brought happiness and joy, while a high VL was described as bad results and would make them feel sad and afraid.

> *When the VL results come back, for those whose results are good, they will be happy. But for those with high VL results, you will find them with closed eyes, very sad and angry. When the results just return from Kampala, those with good VL are smiling all the time but those with bad results close their eyes and keep looking down. There you know there VL is not good.* (49 years old female from Central region)

Participants experienced the outcomes of their viral load test as an indicator of the directions that their health might take, but in their view, it also signaled the social and productive potential of their bodies. While a non-suppressed viral load evoked concern and sympathy about those in this situation, those with supressed viral load had a positive attitude and felt competent to live a meaningful social and productive daily life.

The social importance of VL was discussed in terms of possibilities of a continuation of one's normal social and productive functions. They spoke in terms of ability to interact, to do everything, to be strong, and strength to do things and work very well again.

> *Actually, if you see that the viral load is low, it is actually suppressed, and even those other diseases cannot attack you. And so, you will find that you can live in a strong way and do all your work very well.* (51 years old male from Eastern region)

Some participants described being bothered by the positive feelings and sympathies from their family or friends. They were uncomfortable that their relatives and friends sometimes exaggerated their sympathies, however well-meaning, thereby inadvertently enlisting negative feelings:

> *The experience I got was that I was isolated by the family, community and friends. Not because they hated me or I did something wrong, but they had a feeling that I was just was going to die soon, and they kept on saying, 'That one is just waiting to die.' You know, you live the kind of life where everyone is sympathizing with you.* (42 years old male from Western region)

**Facilitators to VL testing.** Participants identified several factors that influenced them to do VL testing. They indicated that routine counselling and follow-up by the health workers, and peer support, were key in enabling them to decide to do VL testing.

> *My fellow clients facilitate me because whenever we meet here, we discuss a lot of health related issues that concern our lives. Things like the ways we should live, we should not miss taking our drugs, or doing our VL tests, and so on.* (52 years old male from Northern region)

Another important motivation for adhering to VL testing was the desire to avoid inadvertent disclosure of HIV status when one begins to fall sickly and people begin to suspect it, and the desire to return to 'normal' health and reach a point when one can have children while on ART

> *I don't want to fall sick because everyone will get to know that I am HIV positive. My fellow clients also encourage me especially those who have taken ART for long. For example there is a lady who is our neighbour. Before joining TASO, she was almost dead, but right now she is*

*healthy and her children are all healthy. And I am aaaah, still young. I know that I will also have health babies like her.* (22 years old female from Northern region)

Having a previous HIV positive relative or friend who got complications because of not seeking care, and the desire not to disappoint one's caregivers and reduce their caregiving burden also motivated them to take their own drugs and also accept VL testing.

*But if I refuse to take my drugs I will also die very fast like my father. I want to study at least if possible, I be someone [successful] in future. But when I have a high VL, I will be sick every time with diseases like TB and malaria. I will be causing problems again to my mother, aaaah yet she doesn't have money, right now we are four children and she is a single mother hahaha.* (25 years old male from Western region)

**Barriers to VL testing.** Participants reported various barriers which affected VL testing. The main barrier reported was lack of transport money to take them to the health facility for VL testing on the appointed day.

*There are those times when one knows that he/she is due for VL testing appointment, and may be one fails to come at the facility for testing due to lack of transport especially like us women who are taking care of ourselves and also taking care of our big families.* (52 years female from Western region)

Participants reported fearing gossip both at the community and at the facility by other people about their HIV status as barrier to seeking HIV services, including VL testing. While describing facility based gossip, one participant described how the uniqueness of HIV and VL testing betrayed their HIV status since, unlike tests for other diseases which often required little blood taken by simple pricks on a finger, blood drawn for VL testing was usually a lot more and drawn from the arm, raising suspicion that one was doing a major test.

*Stigma from community members is too much and even some times, when eeeh you are seen going for VL because if making other tests which are not HIV tests, they get the blood by use of prickers on the fingers but for the VL, blood is collected from the upper hand. So when they see that a lab attendant is getting a lot of blood from your arm ooooh, that's a problem hahahahaha. . .. People start isolating you when they see you being tested for VL, and they start saying aaah this person is living with this kind of virus and with a lot of allegations, people start giving you space or they limit themselves from involving in any way with you.* (22 years old male from Northern region)

Lack of awareness about the importance of VL testing by some PWH was also mentioned as a barrier. As a result, such people only put their emphasis on taking their drugs but not VL testing. Misconceptions of the purpose of VL testing were also reported, especially among the older PWH, who thought that VL is done to find out whether HIV has been completely cured, and if the result was not what they expected, they lost interest in future testing.

*What I know is that even old people sometimes don't know why they take their blood. Some of them think that maybe, it is to re-check whether they have HIV or not which is true because for most of them, every time they remove their blood, they want to know whether HIV has completely gone or not.* (38 years old female from Eastern region)

Another important factor was conflicting priorities or forgetting the appointment.

*Some people when it's a rainy season like this, they work in their farms and forget their appointment dates. When they go to the garden from morning up to sunset, they think that coming for refill and VL testing is just wasting their time for farming.* (47 years old female from Northern region)

There were many allegations about amount of time spent at the health facility, waiting to do a VL test, and the delayed return or lost VL results as barriers to uptake of VL testing. Participants reported that in some facilities, they are made to wait for long in the queues, and some end up walking away without doing the test. This challenge was attributed to health workers being few in those facilities. Some participants reported that some health workers forget to give appointment dates for VL testing to PWH, hence they do not know when to return to the facility for the test.

*I think the number should be increased because you know, people can easily get tired especially people like us who are living with HIV. I have seen most people come here to receive healthcare services but they sit until they get up and escape back home without receiving healthcare services which includes VL testing. Therefore, the number of healthcare providers should be increased so that all these people can easily be worked on quickly before they get exhausted waiting and go back home.* (35 years old male from Central region)

Describing the alleged delayed return or loss of the VL results, one participant particularly told of the sad experience of another patient who had severally reportedly missed her results for some unknown reason, while others talked about the difficulty of having to return on another date for the results if required to do so.

*They delay to bring our results and, in most cases, it comes late, and other times the results get lost from the lab for good [and never to be traced], and there is a patient I know, she has been complaining that they usually draw her blood to test for her VL and that she has never received the rest results. But I usually tell them that if the results were bad, they would have called them back, but since they have not called you, it means the results were good. And I also encourage her to always ask about it during their clinic visits.* (29 years old female from Western region)

**Required support to enhance VL testing.** The participants made some recommendations on how VL services could be improved. First, they emphasized the need to send continuous reminders to those due for VL testing, and also ensure early return of VL results. Second, participants suggested that VL test results should be returned on the same day. Third, they recommended increased deployment of health workers at health facilities to reduce the waiting time for VL services, and more sensitization, awareness and education to reduce HIV stigma and increase awareness of the benefits of VL testing.

*The health workers should take the initiative to bring the test results in time after taking their blood samples for VL testing so that clients can get to know what has come out of their blood tests to be able to know how they are faring health wise. The number of healthcare providers should be increased so that all these people can easily be worked on quickly before they get exhausted of waiting and go back home without being tested.* (29 years old female from Western region)

Other recommendations were not specific to improving VL testing in particular, but rather aimed at improving the overall support to people living with HIV. They spoke of the need to return to 'a time in the past when things were better for people living with HIV' because there was a wide range of material support, while others believed that government needed to be reminded 'not to forget them'.

*The government used to help people living with HIV. I remember some years ago they distributed soya, they gave them cooking oil, but I don't know what happened to those services.* (31 years old female from Eastern region)

*I wanted to request that let the government see how to help us as people who are living with HIV because we don't have any support. But let the government help us in any way possible and that is what I wanted to say to you. Help us and deliver our report to the government for some support because if you follow it closely, the government seems to have forgotten us.* (45 years old male from Central region)

### Low-level viraemia

**Perceptions and implications of low-level viraemia.** Although all the participants appeared to understand the value of measuring VL and discussed its implications to health, most of them were not aware of the term, 'low-level viraemia.' However many participants with either no education or primary education perceived and interpreted a VL between 50 and 1,000 copies/ml as a decreased and suppressed VL, which they considered to be good and the goal of ART. These participants did not consider this VL range to cause any negative consequences or lead to any poor clinical outcomes.

*With a VL between 50 to 1,000 copies/ml, one can go to the garden and do farming very well. However, if your VL is non-suppressed, you can go to the garden and your head will be spinning and therefore, you cannot do work because your body is weak too but if its lower, you can do everything, come back home from the garden, eat your food and if you want to return to the garden in the afternoon, you can still do, return home after because you are strong.* (47 years old female from Northern region)

The vocabulary of those with LLV test results was often celebratory and several participants described the pleasure, happiness and joy that they experienced. A low-level viral load test result was thus received as a victory–a triumph–that resulted from putting up a great day-to-day adherence effort. Adherence was presented as their day-to-day work, a goal that forwarded their cause and quest for a better life, and was thus worth rejoicing about when it paid off. As a result, achieving LLV required that one subdues any feelings of doubt and fear of defeat as they faced the battle of life on ART. For many, considerable attention was paid to the importance of avoiding worries, and having a positive self-concept in order not to undermine the gains made.

*If you achieve a VL between 50 to 1,000 copies/ml, there will be a lot of joy and thanking God Almighty very much. After realizing that, you don't have to start worrying or doing other things but you have to keep your life very well as you ought to.* (39 years old male from Eastern region)

Interesting to note, is that these participants with either no education or with primary education, believed that they had already developed LLV and were striving to take their treatment

to maintain it. Maintaining LLV was reported as a key motivation for taking ART and routinely doing VL testing, to monitor the health status.

*If the VL copies are low and you follow the advice of the healthcare giver, it means that the VL copies will keep reducing and will not rise up again. For example; when we just received the drugs recently, I had to make sure to keep taking those drugs so that the strength of virus is reduced completely.* (45 years old male from Central region)

A few participants with secondary and tertiary education indicated that a VL between 50 and 1,000 copies/ml is not good, and is caused by not taking drugs very well as prescribed. They said that they desired to always have a non-detectable VL (VL below 50 copies/ml), and they reported this desire as being a key motivator to adhering to their treatment and doing their VL tests routinely.

*Yeah, I have heard about that because I have also experienced it. When I was still in school, I was hiding my drugs and most times I would miss, and when they would do the VL test, I would have some copies like 50, 80, or 100. Yeah. So I know that when some copies are detected, it means that the adherence, it's not really good to what is expected of us, maybe we are missing out on our drugs, maybe we are taking them at the wrong time now and again and that makes us get some copies of the virus detected in blood.* (20 years old female from Western region)

They discussed the resilience of the virus and the imperative to deal with it decisively through proper adherence to reduce it to undetectable levels.

*Because when the copies of the virus remain in blood they can multiply and weaken me, the good thing would be 0 detection of the HIV virus.* (51 years old male from Eastern region)

There were mixed responses among these participants with secondary or tertiary education, with some of them saying that they cannot get LLV if they take their drugs very well, while others indicated that they could still get it.

*I don't think so because I don't miss my drugs and the time I agreed with my health worker is the one I still follow. Maybe if something happens and I don't know really how it has happened but I don't think I will get low-level viraemia.* (20 years old female from Western region)

*Okay that may happen accidentally, because they told us that you may be adhering well to the medication, but still find yourself with some copies of the virus in the blood especially when the medicine fails to work in your blood.* (51 years old male from Eastern region)

## Discussion

In this study, we explored the meaning and perceptions of VL testing and low-level viraemia among PWH on ART in Uganda, using the health belief model. To our knowledge, this is the only such study to be conducted in Uganda, to this date. Our analysis identified seven main themes; local concepts used to describe VL, perceptions of VL testing, implications of VL results, facilitators to VL testing, barriers to VL testing, required support to enhance VL testing, and perceptions and implications of LLV.

The findings indicated that the participants' local descriptions of VL testing nearly matched the medical meaning, and many PWH had mastered the associated medical terms, suggesting

that the participants understood the concept of VL. We found that participants were comfortable describing the numbers associated with VL testing, and their use of the English medical phrases such as suppressed, and non-suppressed was impressive. Our findings are different from the other HIV related studies, which showed that many PWH still remain unaware of some of the HIV services [30, 31]. However, it was reported that few PWH out in the community were unaware of what VL meant, and hence did not seek the service. Basing on this, there is an urgent need to scale up community-based services to sensitize PWH at community level, and to offer them VL services.

On contrary, most PWH had no idea what LLV meant, and others perceived it as having a reduced VL, which is suppressed, and considered good. Interesting to note, many PWH did not think that they were susceptible to LLV while others indicated that they were already in a state of LLV and were so happy to have it since they think that it is the goal of ART. Previous studies have associated residual LLV with reduced drug adherence [32], and hence if PWH do not perceive LLV to be a severe condition, there is less likelihood that they will undertake the different measures like drug adherence and routine VL testing to manage the condition, as stipulated by the health belief model [33, 34].

The PWH noted that lack of transport coupled with poverty was one of the main barriers which hindered them to go to the health facilities for VL testing. A study to understand the role of economic strengthening among adolescents and young adults in Rakai, Uganda also highlighted lack of transport and poverty as key barriers in seeking HIV care [35]. HIV stigma from both the family and community was emphasized as a barrier in seeking VL services, and this has also been highlighted in previous studies both in Uganda and Kenya as a key barrier in HIV care [36, 37]. PWH reported that stigma results into non-disclosure due to fear to be discriminated, which consequently results into both poor drug adherence and fear coming to the health facilities to receive HIV services including VL testing. It was highlighted that some PWH in the community were unaware about VL testing, while others had conflicting priorities, hence not coming for the service. These barriers have also been noted to affect other HIV services in different studies [30, 31]. Delayed VL results return and few numbers of health workers leading to long waiting time at health facilities were key barriers to VL testing, as also highlighted in previous studies for the challenges affecting HIV care [38–40].

Different factors like routine counselling, education and sensitization, family visits by the health workers, family support, community-based sample collection and services and receiving continuous reminders from health workers positively influenced VL testing, as found in other studies on seeking of other HIV services [41–43]. Having a previous HIV positive relative or friend who got complications because of not seeking care was also noted a key motivator for seeking HIV care including VL testing.

Our findings emphasize that there is an inevitable need to institute more strategies like community outreaches to create more awareness about both VL testing and LLV and also manage HIV related stigma at both health facility and community levels. At the present, LLV is perceived as a good outcome of ART and interventions including patient education must be instituted to correct this perception among PWH, and also manage it. There is also need to improve results turnaround time for VL test results by scaling up remote VL results printing in health facilities, increase health workers and motivate them well and also scale up community based VL services.

## Limitations

Our study included only PWH on ART who had gone to the health facilities to seek care, and hence did not include PWH who were in the community. As a result, we might have missed to

hear the perceptions of some of the PWH who do not go to the health facilities to seek care. However we inquired from all the participants what they heard from their colleagues in the community about VL testing and LLV. Hearing from the PWH at community level would have given us a more holistic exploration of the meaning and perceptions of VL testing and LLV among PWH on ART in Uganda. Furthermore, we used the health belief model in our study, and despite its strength, we were not able to understand the attitudes, beliefs and other individual factors which could influence the decision to do VL testing by the PWH, and we also never explored how the decision to do VL testing could be influenced by the habitual behaviours. Finally, our study was a qualitative study whose results are not generalizable but important to note, is that our findings provided an understanding of how our study population perceived LLV and VL.

## Conclusion

Our study findings indicated that many PWH are unaware about LLV, and hence do not understand its associated treatment risks and poor outcomes. Furthermore, many PWH at the health facilities understand VL testing and its benefits, however as reported, some PWH in the communities are not aware of VL testing. Our findings emphasize the imminent need to institute more strategies to create more awareness about both LLV and VL testing, manage HIV related stigma at both health facility and community level, improve results turnaround time for VL test results, increase the numbers of health workers and motivate them well, and scale up community based VL services.

## Supporting information

**S1 Table. Codebook perceptions of VL and LLV study.**
(DOCX)

## Acknowledgments

We extend our acknowledgment to the project administration team of Makerere University behavioral and social science research (MakBSSR) project, including Prof. Moses Kamya, Prof. Anne R Katahoire, Dr. Fred Semitala, Ms. Joanita Nangendo, Mrs. Namubiru Rhoda and Mr. Sempala Moses, for the efforts devoted towards conduction of this study. We also thank Dr. Joseph Matovu, Dr. Odokonyero Raymond, Dr. Charles Patrick Namisi and Mr. Tomusange Joseph for the efforts towards completion of this work.

## Author Contributions

**Conceptualization:** Nicholus Nanyeenya, Godfrey Siu, Noah Kiwanuka, Fredrick Makumbi, Esther Nasuuna, Damalie Nakanjako, Gertrude Nakigozi, Susan Nabadda, Charles Kiyaga, Simon P. S. Kibira.

**Formal analysis:** Nicholus Nanyeenya, Godfrey Siu, Esther Nasuuna, Simon P. S. Kibira.

**Funding acquisition:** Gertrude Nakigozi.

**Investigation:** Nicholus Nanyeenya.

**Methodology:** Simon P. S. Kibira.

**Project administration:** Susan Nabadda, Charles Kiyaga.

**Resources:** Susan Nabadda, Charles Kiyaga.

**Supervision:** Noah Kiwanuka, Fredrick Makumbi, Damalie Nakanjako, Gertrude Nakigozi, Susan Nabadda, Charles Kiyaga.

**Writing – original draft:** Nicholus Nanyeenya, Godfrey Siu, Esther Nasuuna, Simon P. S. Kibira.

**Writing – review & editing:** Nicholus Nanyeenya, Godfrey Siu, Noah Kiwanuka, Fredrick Makumbi, Esther Nasuuna, Damalie Nakanjako, Gertrude Nakigozi, Susan Nabadda, Charles Kiyaga, Simon P. S. Kibira.

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
