## [Decision Letter · Decision Letter 0]

15 Mar 2023

PGPH-D-23-00139

Hopes, Joys and Fears: Meaning and perceptions of viral load testing and low-level viraemia among people on antiretroviral therapy in Uganda: A qualitative study.

Dear Nayeenya,

Thank you for submitting your manuscript to PLOS Global Public Health. After careful consideration, we feel that it has merit but does not fully meet PLOS Global Public Health’s publication criteria as it currently stands. Therefore, we invite you to submit a revised version of the manuscript that addresses the points raised during the review process.

We look forward to receiving your revised manuscript.

Kind regards,

Collins Otieno Asweto, PhD

Academic Editor

Journal Requirements:

2. Your manuscript is missing the following sections: Introduction. Please ensure these are present, and in the correct order, and that any references to subheadings in your main text are correct. An outline of the required sections can be consulted in our submission guidelines here:

https://journals.plos.org/globalpublichealth/s/submission-guidelines#loc-parts-of-a-submission

4. In the online submission form, you indicated that "The codebook for the study has been availed. Any further datasets used and/or analysed during the current study are available from the corresponding author on reasonable request". All PLOS journals now require all data underlying the findings described in their manuscript to be freely available to other researchers, either 1. In a public repository, 2. Within the manuscript itself, or 3. Uploaded as supplementary information.

Reviewers' comments:

Reviewer's Responses to Questions

**Comments to the Author**

1. Does this manuscript meet PLOS Global Public Health’s publication criteria? Is the manuscript technically sound, and do the data support the conclusions? The manuscript must describe methodologically and ethically rigorous research with conclusions that are appropriately drawn based on the data presented.

Reviewer #1: Yes

Reviewer #2: Yes

2. Has the statistical analysis been performed appropriately and rigorously?

Reviewer #1: Yes

Reviewer #2: N/A

3. Have the authors made all data underlying the findings in their manuscript fully available (please refer to the Data Availability Statement at the start of the manuscript PDF file)?

Reviewer #1: Yes

Reviewer #2: Yes

4. Is the manuscript presented in an intelligible fashion and written in standard English?

Reviewer #1: Yes

Reviewer #2: Yes

5. Review Comments to the Author

Reviewer #1: HIV management is one aspect of the field of medicine that has faced a lot of challenges. Initial challenges revolved around acceptance of HIV status, stigma, lack of support systems, and availability of ARVs especially in low and middle-income countries. Emerging challenges are issues to do with adherence to medication and the management of HIV in general. The author's findings on HIV patient perception towards VL testing is a plus in the management of HIV. The ability of PWH to interpret and appreciate the role of LLV in HIV management is a good sign of the acceptability of VL testing by the patients in their management. The conclusion of the study identify areas that needs improvement in the utilization of VL testing in the monitoring and management of HIV patients.

Reviewer #2: Nanyeenya et. al present a qualitative study based on the health belief model in which they assess the perceptions of HIV viral load testing and low-level viremia in people living with HIV in Uganda. This study provides unique insights into patient perceptions and understanding of viral load testing and results interpretation. Such studies are a crucial piece needed for developing more effective interventions to improve the overall care for people with HIV in a manner that centers their needs and enhances their understanding of these interventions. The manuscript is very well written and clearly presented. There are some comments the authors may wish to consider to improve on their work:

Comments:

1. In the discussion I would have liked to see the authors discuss how their results could be used to design interventions which can improve on how patients view low level viremia and even propose plausible interventions that could be effective based on the data they have generated.

2. I applaud the efforts made by the authors to methodically conduct detailed interviews adapted for language and cultural setting which is so crucial to obtain information that accurately represents the views of the respondents however the sample size (n=32 detailed interviews) of the study is quite small which makes one to wonder whether such a small sample size is sufficient to fully capture the full spectrum of perceptions among people living with HIV. Could the authors comment on this choice of sample size and acknowledge this limitation in the manuscript?

3. The authors present their results very well. A summary figure or table of the key themes emerging from the detailed interviews as detailed in the sub-headings in the results section will break the monotony of lengthy text and provide a nice summary of the findings to accompany the detailed reporting of the results.

4. Line 541 – minor typo correct behaviours currently spelled as bahaviors

6. PLOS authors have the option to publish the peer review history of their article (what does this mean?). If published, this will include your full peer review and any attached files.

**Do you want your identity to be public for this peer review?** For information about this choice, including consent withdrawal, please see our Privacy Policy.

Reviewer #1: **Yes: **Festus Mulakoli

Reviewer #2: No

---

## [Decision Letter · Decision Letter 1]

13 Apr 2023

Hopes, Joys and Fears: Meaning and perceptions of viral load testing and low-level viraemia among people on antiretroviral therapy in Uganda: A qualitative study.

PGPH-D-23-00139R1

DearNanyeenya,

We are pleased to inform you that your manuscript 'Hopes, Joys and Fears: Meaning and perceptions of viral load testing and low-level viraemia among people on antiretroviral therapy in Uganda: A qualitative study.' has been provisionally accepted for publication in PLOS Global Public Health.

Best regards,

Collins Otieno Asweto, PhD

Academic Editor

Reviewer Comments (if any, and for reference):

Reviewer's Responses to Questions

**Comments to the Author**

1. If the authors have adequately addressed your comments raised in a previous round of review and you feel that this manuscript is now acceptable for publication, you may indicate that here to bypass the “Comments to the Author” section, enter your conflict of interest statement in the “Confidential to Editor” section, and submit your "Accept" recommendation.

Reviewer #1: All comments have been addressed

Reviewer #2: All comments have been addressed

2. Does this manuscript meet PLOS Global Public Health’s publication criteria? Is the manuscript technically sound, and do the data support the conclusions? The manuscript must describe methodologically and ethically rigorous research with conclusions that are appropriately drawn based on the data presented.

Reviewer #1: Yes

Reviewer #2: Yes

3. Has the statistical analysis been performed appropriately and rigorously?

Reviewer #1: Yes

Reviewer #2: N/A

4. Have the authors made all data underlying the findings in their manuscript fully available (please refer to the Data Availability Statement at the start of the manuscript PDF file)?

Reviewer #1: Yes

Reviewer #2: Yes

5. Is the manuscript presented in an intelligible fashion and written in standard English?

Reviewer #1: Yes

Reviewer #2: Yes

6. Review Comments to the Author

Reviewer #1: I have read through the revised copy of the manuscript and the authors have addressed the initial comments where applicable. I is a good study and it will help in the effective management of HIV patients in Sub-saharan Africa. I will also encourage the authors to extent their scope in future publications for a holistic coverage of topic of study.

Reviewer #2: the authors have adequately addressed the concerns and queries from this reviewer

7. PLOS authors have the option to publish the peer review history of their article (what does this mean?). If published, this will include your full peer review and any attached files.

**Do you want your identity to be public for this peer review?** For information about this choice, including consent withdrawal, please see our Privacy Policy.

Reviewer #1: **Yes: **Festus Mulakoli

Reviewer #2: **Yes: **Boghuma Kabisen Titanji
